# Determinants of Lipid Domain Size

**DOI:** 10.3390/ijms23073502

**Published:** 2022-03-23

**Authors:** Ali Saitov, Maksim A. Kalutsky, Timur R. Galimzyanov, Toma Glasnov, Andreas Horner, Sergey A. Akimov, Peter Pohl

**Affiliations:** 1Institute of Biophysics, Johannes Kepler University Linz, Gruberstraße 40, 4020 Linz, Austria; ali7saitov@googlemail.com (A.S.); andreas.horner@jku.at (A.H.); 2A.N. Frumkin Institute of Physical Chemistry and Electrochemistry, Russian Academy of Sciences, 31/5 Leninskiy Prospekt, 119071 Moscow, Russia; mkalutskiy@inbox.ru (M.A.K.); gal_timur@yahoo.com (T.R.G.); akimov_sergey@mail.ru (S.A.A.); 3Department of Theoretical Physics and Quantum Technologies, National University of Science and Technology “MISiS”, 4 Leninskiy Prospekt, 119049 Moscow, Russia; 4Institute of Chemistry, University of Graz, 8010 Graz, Austria; toma.glasnov@uni-graz.at

**Keywords:** ordered domain, lipid membrane, photoswitchable lipid, domain interaction, theory of elasticity, spontaneous curvature

## Abstract

Lipid domains less than 200 nm in size may form a scaffold, enabling the concerted function of plasma membrane proteins. The size-regulating mechanism is under debate. We tested the hypotheses that large values of spontaneous monolayer curvature are incompatible with micrometer-sized domains. Here, we used the transition of photoswitchable lipids from their cylindrical conformation to a conical conformation to increase the negative curvature of a bilayer-forming lipid mixture. In contrast to the hypothesis, pre-existing micrometer-sized domains did not dissipate in our planar bilayers, as indicated by fluorescence images and domain mobility measurements. Elasticity theory supports the observation by predicting the zero free energy gain for splitting large domains into smaller ones. It also indicates an alternative size-determining mechanism: The cone-shaped photolipids reduce the line tension associated with lipid deformations at the phase boundary and thus slow down the kinetics of domain fusion. The competing influence of two approaching domains on the deformation of the intervening lipids is responsible for the kinetic fusion trap. Our experiments indicate that the resulting local energy barrier may restrict the domain size in a dynamic system.

## 1. Introduction

Phase separation in biological membranes has significant functional implications. Liquid-ordered lipid domains (LODs) and liquid-disordered domains (LDDs) exhibit different permeabilities to small molecules and dissimilar mechanical properties [1,2]. Protein recruitment into liquid-ordered lipid domains provides a scaffolding mechanism [3,4]. In cellular membranes, LODs are called rafts if they are not wider than 200 nm and contain both cholesterol and sphingomyelin [4,5]. Yet, the mechanism that ensures the small domain size and, thus, the required proximity of the active molecules is not entirely resolved. Observations of both LODs and LDDs that are smaller than the diffraction limit in artificial, protein-free lipid bilayers [6,7,8] indicate the existence of a protein-independent mechanism of domain size regulation.

Two main hypotheses have been put forward (Figure 1): According to hypothesis **α**, spontaneous monolayer curvature *J*_0_ is the primary determinant of domain size [9]. In contrast, hypothesis **β** prioritizes line tension, λ [10].

**α** envisions the curvature stress for domains with large *J*_0_ values in compositional symmetric bilayers [11]. The reasoning is that domain registration from the two monolayers results in a flat bilayer (Figure 1). In an asymmetric bilayer fragment consisting of an LDD monolayer on one side and a LOD monolayer on the other side, curvature frustration may be smaller as the resulting bilayer is not necessarily flat. It acquires bilayer curvature *J_B_* [12] if *J*_0_ differs for LDD monolayers and LOD monolayers. In a bilayer with several such fragments, the differently curved areas must alternate [13] to maintain a flat membrane geometry. The division between the two scenarios of symmetric and asymmetric bilayer fragments is not strict, as thermal undulations are supposed to cause transient local asymmetry [14]. In any of the above cases, the domain size decreases with absolute values of *J*_0_. For membrane compositions with large positive *J*_0_ values, the equilibrium domain size is as small as 10 nm [11].

However, **α** is at odds with the observation that domains are always registered, even in asymmetric bilayers [8]. The formation of locally asymmetric fragments is energetically unfavorable because stiffed regions from the two leaflets attract each other [15]. Stiffer lipid domains tend to be distributed into areas with lower monolayer curvature. Consequently, membrane undulations naturally align domains in the opposing monolayers [16]. The corresponding driving force scales with membrane area. It becomes insufficient for the alignment of tiny domains. Accordingly, for smaller domains, another driving force is dominant: λ. It drives LDD and LOD registration [17]. The free energy of registered LODs and LDDs is smaller than that of anti-registered LODs and LDDs [18].

Hypothesis **β** recognizes the vital role of λ for domain formation and registration. λ originates from the hydrophobic mismatch between thinner LDDs and thicker LODs [19,20]. It reflects the energy penalty associated with elastic lipid deformations at the LOD–LDD boundary. Along with this mechanical component, λ also contains a chemical contribution that reflects the difference in lipid concentrations in two contacting phases. The total boundary energy *E_TB_* increases with the interphase boundary’s perimeter [21,22]. As the state of minimal *E_TB_* corresponds to an LOD/LDD system with minimal total boundary length, the domains tend to fuse, giving rise to the macroscopically large domains observed in artificial bilayers. Nevertheless, nanometer-sized domains may still exist because line-active components decrease λ, thereby averting domain fusion [23,24,25].

The goal of the present work is to prove the validity of hypotheses **α** or **β**. Therefore, we change the *J*_0_ of lipids entering the composition of preformed domains by exposure to light. Domain dissipation due to increased absolute *J*_0_ values would prove hypothesis **α**. Light-triggered conformational lipid changes may also affect λ, provided that the photo-sensitive lipid adopts a localization at the phase boundary or modifies the elastic properties of bulk phases. For this case, hypothesis **β** predicts changes in domain size evolution.

## 2. Results

We formed membranes across the aperture of a Teflon septum. To enable the development of lipid domains, we kept close to the so-called “canonical” raft-forming mixture 1:1:1 DOPC:DPPC:Chol [6] by substituting (i) DOPC for DPhPC [26] and (ii) half of DPPC for the photoswitchable lipid, also called photolipid, PhoDAG-1 [8].

Illumination at a wavelength of 474 nm switched the conformation of PhoDAG’s photo-sensitive acyl chain from the *cis*-state into the *trans*-state, i.e., the cone-shaped PhoDAG-1 transformed into an almost cylindrical lipid, thereby decreasing the absolute value of *J*_0_ [27,28] (Figure 2).

According to hypothesis **α**, this intervention should allow LDDs to increase in size. However, many LDDs dissolved (Figure 3). We used fluorescence laser scanning microscopy for visualization (see Section 4). Lipid domains of all sizes were visible as the dyes partitioned preferentially into LDDs. Domain tracing enabled us to calculate the diffusion coefficients of LODs and LDDs, allowing us to assign domain size [8]. The method works well also for domains that are smaller than the diffraction limit.

To calculate weighted concentration averages of the lipid components’ *J*_0_ (Table 1), we utilized the phase diagram for DPhPC:DPPC:Chol mixtures [29] and assumed (i) *trans*-PhoDAG-1′s packing density to be similar to DPPC, (ii) *cis*-PhoDAG-1′s packing density to be similar to DPhPC, and (iii) equal distribution coefficients between LODs and LDDs for (a) *trans*-PhoDAG-1 and DPPC and (b) *cis*-PhoDAG-1 and DPhPC. We calculated the following compositions of LODs: 3.5:28.5:39.5:28.5 DPhPC:DPPC:Chol:DAG in PhoDAG-1′s *trans*-state; 5:39:54:2 DPhPC:DPPC:Chol:DAG in PhoDAG-1′s *cis*-state. We arrived at the following compositions of LDDs: 53:9:29:9 DPhPC:DPPC:Chol:DAG in PhoDAG-1′s *trans*-state; 45:8:25:22 DPhPC:DPPC:Chol:DAG in PhoDAG-1′s *cis*-state.

The number of LDDs increased again upon photoswitching PhoDAG-1 back to its *cis*-state (Figure 3). Most domains exceeded 200 nm in size. It is important to note that we would have detected LODs smaller than 100 nm by virtue of their size-characteristic diffusion coefficient [8]. The observation of large LDDs is in sharp contrast to hypothesis **α**, which predicts a decrease in domain size for increasing absolute *J*_0_ values (compare Figure 1). Photoswitching produced only minute *J*_0_ changes for LODs (Table 1). Accordingly, the failure of the bulky LOD phase to dissipate into smaller LOD domains cannot be used to refute or confirm **α**’s predictions. 

Yet, the experimental observation (Figure 3) is in line with hypothesis **β**. The hydrophobic thickness of the longer *trans*-PhoDAG-1 [27] matches that of DPPC lipids, while the hydrophobic thickness of the shorter *cis*-PhoDAG-1 [27] matches that of DPhPC lipids. Accordingly, **β** predicts the longer *trans* lipids’ assembly into LODs, and the ordered phase’s total area increase upon *cis*-to-*trans* photoswitching (Figure 3). For the same reason, *trans*-to-*cis* photoswitching drives the LDD assembly within macroscopic LODs (Figure 3).

Hypotheses **α** and **β** fundamentally differ in their predictions about the time-dependence of domain sizes: **α** envisages a curvature frustration-driven dissolution of large LDDs. At the same time, **β** foresees continuous domain growth driven by the tendency to minimize the total boundary energy *E_TB_*. We were able to observe micrometer-wide, i.e., macroscopically large, LDDs that did not dissolve after we switched the photo lipid to its *cis* state (Figure 4). This observation disproved hypothesis **α**.

In contrast—as predicted by hypothesis β—LOD growth occurred, albeit very slowly (Figure 4). LDD growth was even slower. A thorough analysis was required to confirm it (Table 1). As domain fusion appeared as the primary growth mechanism, we may quantify the domain growth rates by the rates of merger (Table 1). To evaluate the impact of *cis*-PhoDAG-1 on fusion, we followed 10 LODs and 20 LDDs. We counted events that diminished the number of domains as mergers. We registered those collisions that did not reduce the number of domains as bouncing events. The maximum tracking time was equal to 50 s. The analysis did not include the first 25 s after photo-switching, as the low frame rate of one image per 1.7 s hinders tracing the initially small domains [8]. Both small and large *cis*-PhoDAG-1-containing LDDs bounced from each other in 80% of all collisions. Consequently, small LDDs survived for a prolonged period (Figure 4). In contrast, LODs with *cis*-PhoDAG-1 merged more frequently—after 47% of all collisions.

Next, we tested the hypothesis that the domain growth rate depends on the photo lipid’s conformation. Therefore, we repeated the experiments (Figure 4) after switching the photolipid into its *trans*-state. The analysis of *trans*-PhoDAG-1′s effect on fusion included 20 LODs and 20 LDDs. It abided by the above-applied rules for *cis*-PhoDAG-1. Interestingly, both LDDs and LODs showed continued growth due to mergers and collisions (Figure 5). We were able to discern unsuccessful domain collisions and successful fusion events. Domain merger and bounce can be considered as a discrete random process, the result of which is either merger with probability *p* or bounce with probability (1 − *p*). Assuming a normal distribution for a large number of events *N*, it allows estimating the uncertainty of the merger probability, *p* (Table 1).

Notably, the success rate was size-dependent. The collisions between smaller LDDs and between smaller LODs had a higher probability of success. Larger *trans*-PhoDAG-1-containing domains bounced from each other in 50% of all collisions (Figure 6).

The overall merger rate in the case of *trans*-PhoDAG-1 was 2–4 times higher than in the case of *cis*-PhoDAG-1 (Table 1). Moreover, there was an apparent difference between the dynamics of small and large domains: the efficiency of collisions between large domains lagged behind that of collisions between small domains. The number of small domains decreased faster, thus manifesting a more significant fraction of merging events. The observation suggests a size dependence of the fusion-opposing energy barrier. To clarify whether hypothesis **β** may provide a rationale for both observations—the dependence on (i) the state of PhoDAG-1 and (ii) domain size—we performed a theoretical analysis based on membrane elasticity theory [30] and the corresponding theoretical framework [17,31].

We calculated the dependence of the membrane’s elastic energy on the distance between the boundaries of the domains. In our calculation (see Materials and Methods) of the lipid domains’ interaction energy profiles (Figure 7), we accounted for bending, tilt, and compression/stretching deformation modes characterized by the corresponding elastic moduli: *B*, *K_t_*, and *K_a_*. The energy required for bending depends on *J*_0_. A lipid reservoir (torus) subjects the membrane to lateral tension, 2*σ*_0_. We minimized elastic energy with respect to deformation fields at a fixed distance between the domain boundaries for a discreet set of distances.

Cholesterol is essential for lipid domain formation in our experiments. Its exact role in domain fusion is less clear. Nevertheless, it is likely that lipids with high values of spontaneous curvature dominated domain interaction energy profiles (Figure 7a). As only *cis*-PhoDAG-1 has a higher negative curvature than cholesterol, it appears plausible that cholesterol’s effect on the LOD–LOD and LDD–LDD interactions was more pronounced when PhoDAG-1 is in a *trans*-state than in *cis*-state.

Calculating *E_TB_* as a function of the distance between domain boundaries showed a maximum *E_B_* at a distance *D* ~4–6 nm (Figure 7a). *E_B_* resulted from the mechanical work bestowed on the lipids between two approaching domains (Figure 7b). That is, the intervening lipids departed from the conformation they had adjacent to an isolated domain to adapt to their new environment that may differ in hydrophobic thickness, as well as in splay and tilt of the adjacent lipids. As their initial state served to maintain the energetic minimum for the isolated domain, the new conformation must represent a state of increased energy.

For two *trans*-PhoDAG-1-containing LODs to approach each other, i.e., to shorten the distance *D* between their edges from ~7 nm to ~3 nm, the LODs must overcome the barrier *E_BO_* = 0.03 *k_B_T* per 1 nm of contact boundary length. To obtain energy barrier *E_i_* that opposes domain mergers, the specific energy *E_B_* is multiplied by the size of the boundary *l_R_* involved in domain repulsion. For circular domains, one may use the Derjaguin approximation [32], according to which the effective length of the domain boundary entangled in the interaction is $l_{R}=2\sqrt{2l_{c}R_{0}}$, where *l_c_* is the characteristic decay length of deformation (~1 nm) and *R*_0_ is the domain radius. The Boltzmann factor $e^{-E_{i}/k_{B}T}=e^{-l_{R}E_{B}/k_{B}T}$ governs the probability of domain mergers. The prefactor *A* accounts for the frequency of the merging attempts. Thus, we can write Equation (1) for the logarithm of the merger probability *p*:(1)$$\ln\left(p\right)=-l_{R}\frac{E_{B}}{k_{B}T}+\ln\left(A\right)$$

We approximated *E_i_* and *E_B_*, as well as the probabilities of domain fusion events (Table 1), assuming a constant frequency of fusion attempts, *A*. The assumption seems justified as the *A* governing the domain diffusion coefficient depends but weakly on domain radius [33]. Similarly, the other *A*-determining factor, the characteristic frequency of thermal domain boundary fluctuations, is independent of the domain size, as governed by membrane viscosity. *l_R_* is roughly equal to 8 nm for domains of ~8 nm in radius according to the Derjaguin approximation. For micrometer-sized domains, *l_R_* increased to about 60 nm, yielding *E_i_* values of several *k_B_T*. In contrast, the experimentally observed ratios of fusion probabilities *p*_cis_/*p*_trans_ were close to unity (1.2–2.8, Table 1), suggesting that only a tiny part of any domain boundary interacts with the border of the opposite domain at any time, i.e., *l_R_* ≈ 8 nm works for domains of all sizes.

We found *E_io_* ≈ 0.24 *k_B_T* for *trans*-PhoDAG-1-containing LODs. Similar calculations for LDDs yielded *E_Bd_* = 0.02 *k_B_T*/nm (*E**_id_* ≈ 0.16 *k_B_T*), which is compatible with the observed probability of domain mergers (Table 1). For *cis*-PhoDAG-1, the energy barriers for LOD–LOD and LDD–LDD interactions were equal to *E_Bo_* = 0.06 *k_B_T*/nm and *E_Bd_* = 0.16 *k_B_T*/nm, respectively. The corresponding total interaction energies were *E_io_* = 0.47 *k_B_T* and *E_id_* = 1.26 *k_B_T* (Table 1).

## 3. Discussion

Our experiments unambiguously refuted the hypothesis about *J*_0_′s domain-size-governing role. The observed domain diameters always spanned an extensive range, varying from ~100 nm to 10 s of µm. Curvature stress due to high *J*_0_ values did not lead to domain dissipation. This observation indicates that curvature stress is either independent of domain radius or negligibly small. A simple model for the energy density *F* of a membrane corroborated the conclusion (Equation (2)). It describes *F* as depending quadratically on *J*_B_ and local principal membrane curvatures *C*_1_ and *C*_2_:(2)$$F=\frac{\kappa_{b}}{2}{\left(C_{1}+C_{2}-J_{B}\right)}^{2}+\kappa_{G}C_{1}C_{2}$$
where *κ_G_* is the Gaussian rigidity. In flat (*C*_1_ = *C*_2_ = 0) symmetric bilayers, *F* is equal to zero as the curving propensities of the individual monolayers counteract each other, i.e., *J*_B_ is equal to zero. 

Focusing on the individual monolayers does not change the result. Assuming that the contribution of the individual monolayers to *F* is additive yields Equation (3) for the free energy density *F_M_* of a flat monolayer:(3)$$F_{M}=\frac{\kappa_{b}}{4}J_{0}{}^{2}$$

The total energy *E_M_* is a function of total domain area *A_D_*:(4)$$E_{M}=\frac{1}{4}A_{D}\kappa_{b}J_{0}{}^{2}$$

According to Equation (4) *E_M_* does not increase or decrease if a large domain dissipates into small ones, as long as *A_D_* remains invariant.

The question arises about how molecular dynamics simulation may have corroborated hypotheses **α** [11]. The following assumptions could be responsible: (i) LDDs and LODs have similar elastic parameters, and (ii) LODs have significant positive *J*_0_ values. However, assumption (i) artificially decreases *E_TB_*. Assumption (ii) is invalid because naturally occurring LODs generally have negative or only slightly positive *J*_0_ values. Cholesterol’s *J*_0_ amounts to −0.37 nm^−1^ [34], DPPC’s reaches +0.1 nm^−1^ [35], and DOPC’s or POPC’s *J*_0_ equals −0.091 nm^−1^ or −0.022 nm^−1^, respectively [36]. Attaining the assumed *J*_0_ values of +2 nm^−1^ would have required membrane-destabilizing amounts of short-tail lysophosphatidylcholine [37].

It has been proposed that the sign of *J*_0_ does not matter, as curvature frustration per se may be sufficient for **α** to work [12]. Yet, vanishing line tension at the domain boundary was the cornerstone of hypotheses **α**, and it is unclear how negatively curved monolayers may produce such negligible λ values. Only positive *J*_0_ values may decrease the hydrophobic mismatch at the LOD–LDD boundary (Figure 1). On the contrary, two negatively curved monolayers’ hydrophobic interaction compresses the lipids at the midpoint of the arches (Figure 8). The hydrophobic thicknesses at the domain interface remain unaltered. The persisting hydrophobic mismatch between LODs and LDDs incurs considerable λ values, the minimization of which will govern domain size. Thus, we conclude that the sign of *J*_0_ matters. Realistic *J*_0_ values do not support the idea of curvature-controlled domain sizes.

Our experiments thus indicate that *λ* (and not *J*_0_) governs domain size. As the energy stored in the bilayer’s rim decreases with the border’s length, domains tend to fuse. For two domains to approach, the intervening lipids must be released from the strain that tilts or bends them (Figure 7b). Accordingly, this pre-existing lipid deformation may give rise to a kinetic trap that opposes neighboring domains from coming close to each other. It is caused by a local energy maximum, *E_B_*, that can be calculated as a function of the distance between domain boundaries (Figure 7a). Our calculations predicted that *E_B_* for two interacting LODs might differ from *E_B_* for interacting LDDs. The prediction has been confirmed by experimental observations (Table 1).

In our experiments, we utilized a photoswitchable lipid. PhoDAG-1′s effect on membrane organization is remarkable as its regulatory effect takes hold (i) instantly and (ii) without otherwise perturbing the membrane. As induced by a mere shape change of the molecule, various other switchable molecules may exert similar effects. That is, molecules may be devised to regulate domain size distribution and interaction specifically. Such tools could be used to manipulate membrane lipid and protein distribution between LODs and LDDs. Conceivably, their applicability extends to controlling membrane protein activity.

Line-active substances such as PhoDAG-1 enhance the kinetic trap hampering the approach of neighboring domains. Various other line-active substances may exert similar effects. Producing them may enable cells to manipulate the membrane lipid and protein distribution between LODs and LDDs or alter membrane trafficking [38,39]. For example, ganglioside GM_1_ notably alters raft morphology even when present in fractions of a molar percent [24,25].

## 4. Materials and Methods

### 4.1. Materials

1,2-diphytanoyl-*sn*-glycero-3-phosphocholine (DPhPC, 16:0(3me,7me,11me,15me)), 1,2-dipalmitoyl-*sn*-glycero-3-phosphocholine (DPPC, C16), and cholesterol were purchased from Avanti Polar Lipids, Inc. (Alabaster, AL, USA). The photoswitchable ceramide PhoDAG-1 [27] was synthesized as previously described [27]. Labeled phospholipids Atto565 DPPE (1,2-dipalmitoyl-*sn*-glycero-3-phosphoethanolamine, C16) and Atto633 PPE (1-palmitoyl-2-hydroxy-*sn*-glycero-3-phosphoethanolamine, C16) were obtained from ATTO TEC GmbH (Siegen, Germany) and used for visual differentiation of the two leaflets. The aqueous solution contained 20 mM HEPES and 20 mM KCl (pH = 7.0). Hexane and hexadecane were from Merck (Darmstadt, Germany).

### 4.2. Planar Membrane Formation

The two chambers were first filled with preheated buffer solutions. They were separated by a Teflon septum, the aperture of which was pre-treated with 0.5% hexadecane in hexane (volume ratio 1:199) and allowed to dry before filling the chamber [40]. On top of the solutions, homogeneous monolayers were formed from a 2:1:2:1 mixture of DPhPC:DPPC:Chol:PhoDAG-1 in hexane (10 mg/mL). The Atto565-DPPE dye was added only to the *cis*-side (0.004 mol%); the Atto633-PPE dye was added only to the *trans*-side (0.004 mol%). The asymmetric planar membranes were formed by lowering the aperture beneath the monolayers’ level [41]. The experiments were performed after the solutions had cooled to room temperature (22 °C). Membrane quality was controlled by conductance and capacitance measurements, yielding values < 10^−8^ S cm^−2^ and ≥0.7 µF/cm², respectively. The overlay of the fluorescence excited at 561 nm and 633 nm showed domain registration (Figure 3, Figure 4, Figure 5 and Figure 6).

### 4.3. Light Scanning Microscopy (LSM) and Polychrome V Monochromator

The membrane was scanned using a laser scanning microscope (LSM 510 META, Carl Zeiss, Jena, Germany). Atto565-DPPE and Atto633-PPE were excited at 561 nm and 633 nm, respectively. We used a Polychrome V monochromator (TILL Photonics, Gräfelfing, Germany) to switch PhDAG-1 from its *trans* to its *cis* state by illumination at 365 nm and back by illumination at 475 nm light [28].

### 4.4. Theoretical Model

The theoretical framework was described previously [17,31]. In brief, we used the elasticity theory of liquid crystals adapted to lipid membranes [30] to calculate lipid domains’ interaction energy profiles. It describes anisotropic lipid molecules’ average orientation by a vector field of unit vectors called directors, **n**. The field of directors is set at some surface lying inside the lipid monolayer, referred to as the dividing surface. The shape of the dividing surface is described by a vector field **N** of the unit normal to it. We take into account the following deformations of the membrane: (1) splay, characterized by splay modulus, *B*, and by the divergence of the director along the dividing surface, div(**n**); (2) tilt, characterized by tilt modulus, *K_t_*, and by tilt-vector **t**, equal to the deviation in the director from the normal at a given point of the dividing surface, **t** = **n** − **N**; (3) lateral compression/stretching, characterized by stretching modulus, *K_a_*, and by the relative change in the area of the dividing surface, *α* = (*a* − *a*_0_)/*a*_0_, where *a* and *a*_0_ are current and initial areas per molecule at the dividing surface, respectively. The membrane may be subjected to lateral tension, 2*σ*_0_, set by a lipid reservoir. The deformations are set at the specific dividing surface, at which the splay and the lateral compression/stretching deformations are energetically decoupled. This surface is referred to as the neutral surface; it was shown to lie in the region of the junction of polar heads and hydrophobic tails [42]. We assumed that the domain radius significantly exceeds the characteristic length of deformation decay, which usually amounts to several nanometers [17,22,31]. In the case of a translational symmetry along the domain boundary, the system becomes effectively unidimensional. As a result, Gaussian curvature makes zero contribution to the elastic energy.

We introduced a Cartesian coordinate system *Oxyz*. The origin *O* is located in the middle between the boundaries of two interacting domains: the *Oy* axis is parallel to the boundaries, the *Ox* axis is perpendicular to the boundaries, and the *Oz* axis is perpendicular to the membrane plane. The distance to the *Oxy* plane describes the shapes of the neutral surfaces of the deformed monolayers. The deformations are assumed small, and the energy is calculated up to the second order. The thus approximated elastic energy functional may be written as follows:(5)$$W=\int dS\left[\frac{B}{2}{\left(\mathrm{div}\left(n\right)+J_{0}\right)}^{2}-\frac{B}{2}J_{0}^{2}+\frac{K_{t}}{2}t^{2}+\frac{K_{t}}{2}{\left(\alpha -\alpha_{0}\right)}^{2}+\frac{\sigma_{0}}{2}{\left(grad\left(H\right)\right)}^{2}\right],$$
where *J*_0_ is the spontaneous curvature of the lipid monolayer, *H* is the distance from the reference plane to the monolayer’s neutral surface measured along the normal to the reference plane, and *α*_0_ = *σ*/*K_a_* is the equilibrium stretching of the monolayer caused by the lateral tension. The membrane is assumed flat and parallel to the *Oxy* plane, far away from the boundary. The elastic energy functional, Equation (5), was minimized with respect to deformation fields at a fixed distance between the domain boundaries for a discreet set of distances. The resulting dependence of the energy on the distance between domains gives the interaction energy profile.

### 4.5. Parameters of the System

We calculated the energy profile of the mutual interaction of ordered and disordered domains for PhoDAG-1 lipid in *cis* and *trans* states. For the graphical illustration of the obtained results, we used the following parameters (indices “*o*” and “*d*” denote the ordered and disordered domains, respectively): *B*_o_ = 20 *k_B_T* (*k_B_T* ~4 × 10^−21^ J), *B_d_* = 10 *k_B_T*, *K_a_* = 30 *k_B_T*/nm^2^ = 120 mN/m [43], *σ* = 0.1 *k_B_T*/nm^2^ = 0.4 mN/m, *h_d_* = 1.3 nm, *h*_o_ = 1.8 nm. We used different elastic moduli for LODs and LDDs as their splay moduli may differ 2–5 fold [44]. In contrast, the tilt modulus is almost independent of the membrane composition [30]. Generally, *J*_0_ depends on the composition (and correspondingly on the particular membrane domain). We calculated *J*_0_ values (Table 1) as the weighted concentration average of the components’ *J*_0_ using the phase diagram of DPPC/DPhPC/Chol [29]: *J*_chol_ = −0.37 nm^−1^ [34]; *J*_DPPC_ = 0.07 nm^−1^ [35]; *J*_DPhPC_ = −0.2 nm^−1^; *J*_DAG-*trans*_ = −0.2 nm^−1^; *J*_DAG-*cis*_ = −1 nm^−1^ [28].

The assumption that *J*_0_ of a lipid mixture may be calculated as the weighted concentration average of the components’ *J*_0_ is commonly accepted. It has been used to determine most values of individual lipid’s (DPPC, DPhPC, cholesterol, etc.) spontaneous curvatures available today, as the underlying experiments were conducted in binary lipid mixtures [36,37]. Molecular dynamics simulations arguing for nonadditive compositional curvature energetics of lipid bilayers [35] have thus far not been confirmed experimentally. Importantly, molecular dynamics only allow the calculation of the product formed from lipid mixture’s average spontaneous curvature and average bending modulus. They cannot provide values for the spontaneous curvature as such [35]. 

The elastic parameters (moduli of elasticity, spontaneous curvature, etc.) are experimentally determined with a typical error of ~10–15% (e.g., [36,43]). Accordingly, our calculations of the energy barriers and average spontaneous curvatures of the coexisting lipid phases contained the same error plus the uncertainties of the lipid composition born by the phase diagram of the lipid mixture. Given this uncertainty, the *J*_0_ values (Table 1) of LODs for *cis*-PhoDAG-1 (−0.2 nm^−1^) and *trans*-PhoDAG-1 (−0.26 nm^−1^), as well as of LDD for *trans*-PhoDAG-1 (−0.25 nm^−1^), did not significantly differ from each other, whereas *J*_0_ of LDD for *cis*-PhoDAG-1 (−0.4 nm^−1^) differed significantly from the other *J*_0_ values.

## Figures and Tables

**Figure 1 ijms-23-03502-f001:**
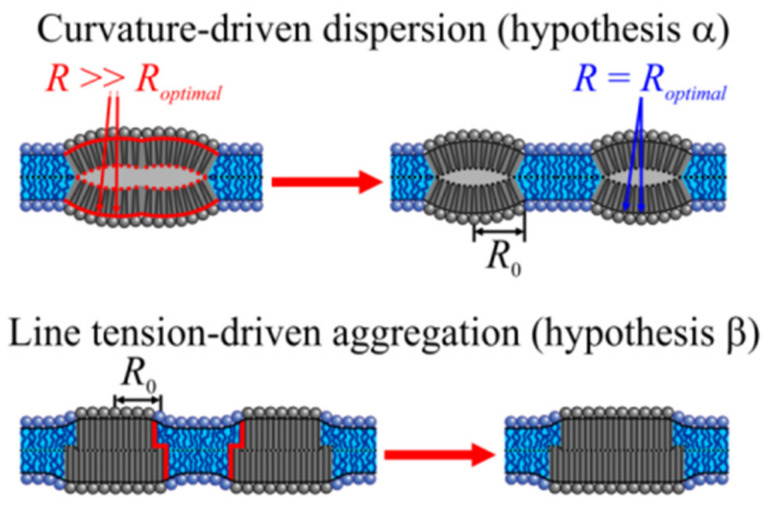
Determinants of lipid domain size. According to alternative hypotheses **α** and **β**, the equilibrium domain size is either governed by spontaneous monolayer curvature, *J*_0_ = 1/*R_optimal_* (top), or line tension, λ (bottom). **α**. As the bending propensities of the upper and lower monolayers are identical, the resulting LOD is nearly flat. That is, the radius *R* of the geometrical curvature of its monolayer surface is much larger than the radius, *R_optimal_*, corresponding to the spontaneous monolayer curvature. Dissipation of the large domain into smaller domains of equal total area releases the curvature stress. *R_optimal_* governs the radius *R*_0_ of the new LODs. **β**. LOD–LOD fusion releases part of the energy spent to deform lipids at the height-mismatched LOD/LDD boundary (red line) as boundary length decreases. The gain in energy scales with line tension λ. Accordingly, λ determines the rate of fusion and thus domain size in a nonequilibrated system.

**Figure 2 ijms-23-03502-f002:**
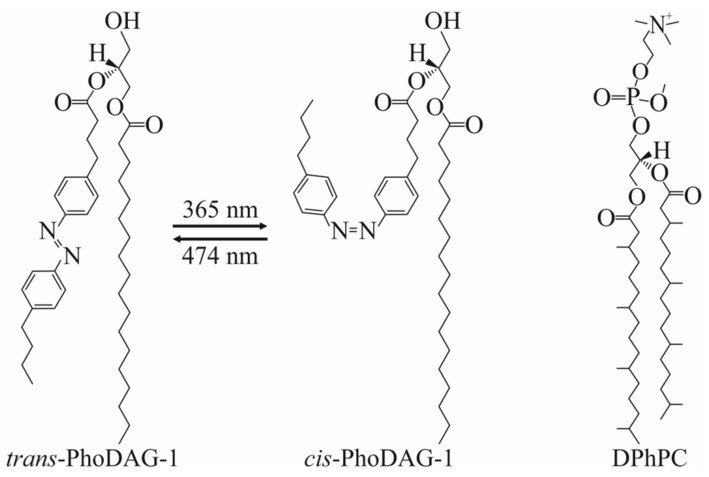
Photolipids and branched lipids. The photoswitchable PhoDAG-1 has two conformations: Left-*trans*-isomer and middle-*cis*-isomer. Illumination at 365 nm induces azobenzene’s *tr**ans*-to-*cis* isomerization; *cis*-to-*trans* isomerization occurs upon illumination at 474 nm. The right panel shows the chemical structure of DPhPC.

**Figure 3 ijms-23-03502-f003:**
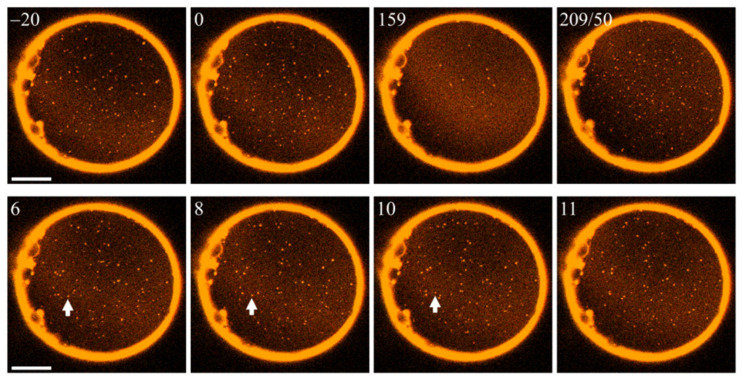
Photoswitching PhoDAG-1 affects phase separation. Illumination at 474 nm (top row, left three frames) dissolves LDDs (bright spots). The numbers in the left upper corners of the images indicate the time (in seconds) after setting the wavelength to 474 nm. The wavelength change at *t* = 159 s to 365 nm results in LDD reassembly. The 2nd number in the fourth frame of the top row indicates how long we illuminated the membrane at 365 nm. The bottom row illustrates the gradual dissolution of a particular domain (marked by a white arrow). The scale bar has a length of 50 µm. The experiment was carried out at an elevated temperature of 30 °C.

**Figure 4 ijms-23-03502-f004:**
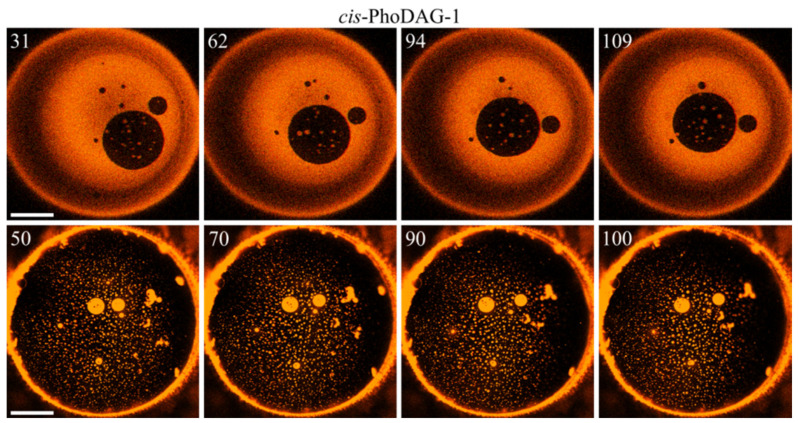
Domain time evolution. **Top row**: The bulky and bright LDD domain covers most of the membrane surface. Due to its content of *cis*-PhoDAG-1, *J*_0_ is very high. Nevertheless, the LDD does not dissipate into small LDDs. The large LOD (large dark circle in the fourth quadrant) grows only ~2% between the first and the last image, indicating delayed LOD–LOD fusion. **Bottom row**: the bulk of the membrane lipids are in the ordered state. The diameter (>10 µm) of the two visible bright LDDs in the center does not change between the first and last image, indicating a hampered LDD–LDD merger. The same conclusion can be drawn from the minor decrease in the total domain number (in an arbitrarily chosen square 25 × 25 μm in the center of the images of the bottom row, the number of domains was 18, 18, 17, 17 from left to right frame). The time (in seconds) after photoinducing *cis*-PhoDAG-1 is in the left upper corner of the images. The scale bar has a length of 50 µm.

**Figure 5 ijms-23-03502-f005:**
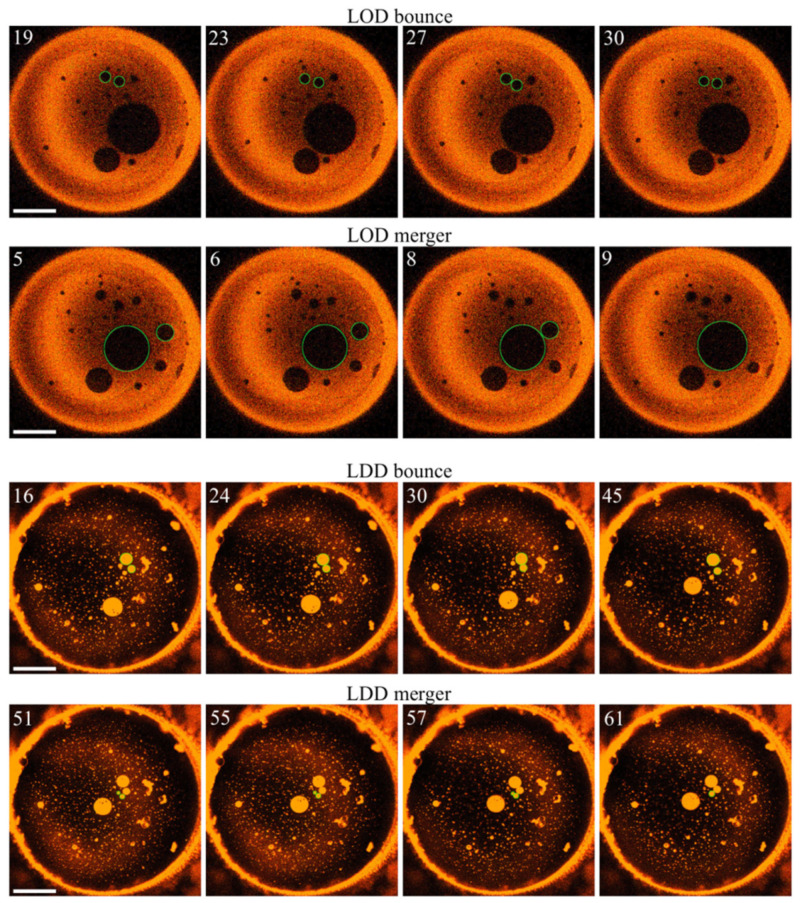
Size evolution of LODs and LDDs. The top two rows reflect representative events, eventually leading to LOD growth. Unsuccessful fusion events (domain bouncing) and successful fusion events are discernible, as depicted in the first and the second rows. The third and fourth rows show attempted LDD fusion events (bouncing) and a successful LDD merger (bottom row), respectively. Green circles mark the domains of interest. The numbers in the left upper corners of the images indicate the time (in seconds) after setting the wavelength to 474 nm, i.e., all pictures were obtained for *trans*-PhoDAG-1. The scale bar has a length of 50 µm.

**Figure 6 ijms-23-03502-f006:**
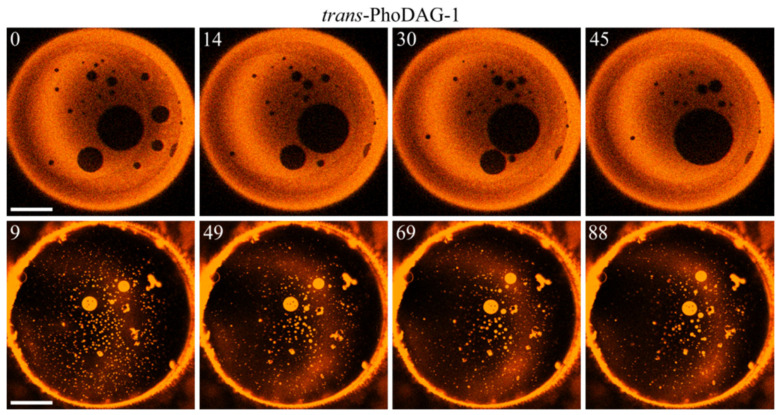
The number of *trans*-PhoDAG-1-containing domains decreases with time. Small LODs (dark, top row) and LDDs (light, bottom row) easily merge, which results in a fast decrease in domain number at an approximately constant area of the corresponding phase. The time (in seconds) after photoinducing *trans*-PhoDAG-1 is in the left upper corner of the images. The scale bar has a length of 50 µm.

**Figure 7 ijms-23-03502-f007:**
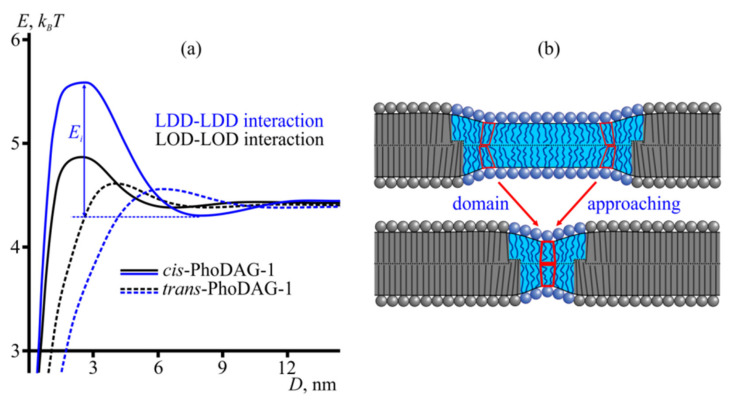
Membrane-mediated domain interaction. (**a**) Energy profiles of domain interactions in the presence of *trans*-PhoDAG-1 (dashed lines) and *cis*-PhoDAG-1 (solid lines). Black and blue plots present the interaction energy (elastic energy of the membrane) as a function of the distance between two LODs and two LDDs, respectively. The energy barrier toward domain fusion, *E_i_*, is illustrated. (**b**) The energy barrier to domain fusion arises from the competition between two domain boundaries to orient the intervening lipids. The lipids at a certain distance from the domain boundary were tilted and bent (top) to minimize the energy stored in an isolated LOD boundary. However, an adjacent second domain forces them to depart from that energy-minimizing orientation (bottom). The new, nontilted orientation minimizes the total elastic energy for the system consisting of two LODs. Yet, it signifies increased *E_TB_* for each of the two LODs. Thus, elastic deformations may kinetically stabilize nanodomains.

**Figure 8 ijms-23-03502-f008:**
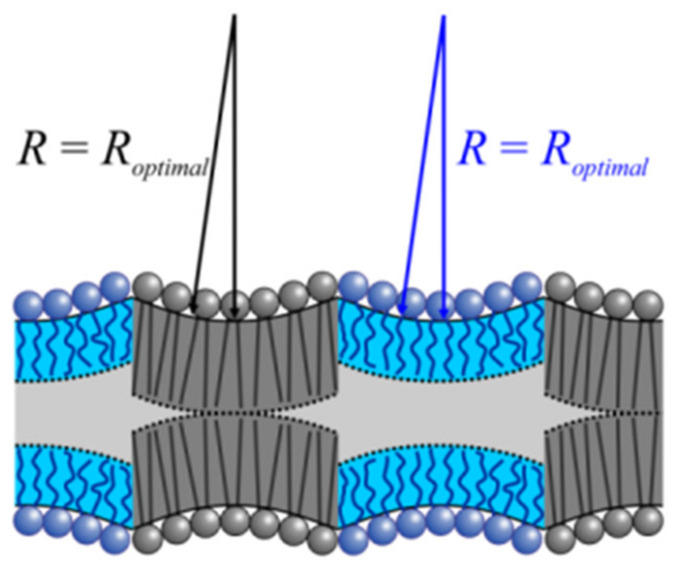
Negative monolayer curvature cannot govern LOD size. The mutual attraction of two registered LODs would lead to the compression of the lipids at the midpoint of the arches instead of decreasing the hydrophobic mismatch at the domains’ edges. Nonnegligible line tension is the consequence. The scheme assumes that the actual inverse monolayer curvature 1/*J*_0_ is close to optimal radius *R*.

**Table 1 ijms-23-03502-t001:** PhoDAG 1′s effect on the fusion of large domains, the fusion-opposing energy barrier *E_i_*, the fusion-opposing energy barrier per unit length of domain boundary, *E_b_*, the probability of merger, *p*, and the calculated spontaneous curvatures, *J*_0_. *N* is the number of events; *A* is the frequency of merger attempts (see Equation (1)).

PhoDAG-1 State	*trans*	*cis*
**LODs**		
*N*, merged	30	9
*N*, bounced	19	10
Merging rate, events/s	30/200 = 0.09	9/210 = 0.025
*p*	61 ± 14%	47 ± 22%
*E_b_*, *k_B_T*/nm	0.03	0.06
*E_i_*, *k_B_T*	0.24	0.47
*k_B_T* × (ln(*A*) − ln(*p*)), *k_B_T*	0.16 ± 0.2	0.43 ± 0.4
*J*_0_, nm^−1^	−0.26	−0.2
**LDDs**		
*N*, merged	53	21
*N*, bounced	42	86
Merging rate, events/s	53/80 = 0.4	21/60 = 0.2
*p*	56 ± 10%	20 ± 8%
*E_b_*, *k_B_T*/nm	0.02	0.16
*E_i_*, *k_B_T*	0.16	1.26
*k_B_T* × (ln(*A*) − ln(*p*)), *k_B_T*	0.25 ± 0.17	1.27 ± 0.34
*J*_0_, nm^−1^	−0.25	−0.4

## Data Availability

Data available on request from peter.pohl@jku.at.

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
