# Peer review of "Determinants of Lipid Domain Size"

_ijms, 2022, doi:10.3390/ijms23073502_

Round 1

Reviewer 1 Report

The authors explore the main reasons for domain sizes. The experimental approach is interesting, however, the manuscript is very difficult to read. Authors should rewrite the manuscript to make it more understandable (and for a broader audience), in order to publish it.
As is, I am not sure to completely understand it, and thus, I am not sure about the quality.

Some comments:
Abstract, line 21:  photolipids is only used in the abstract, afterwards, photo lipids is used. Unify. Besides, it is not a broadly known word, should be defined.

Figs 1 and 7: the grey region between hemilayers does not make sense… molecules deform to avoid such empty spaces. Please improve the schemes

Lines 92-95: Add reference for sentence “As photoswitchable lipid, we used diacylglycerol (PhoDAG-1). Illumination at a wavelength of 365 nm switched the conformation of PhoDAG's photo-sensitive acyl chain from the cis-state into the trans-state, i.e., the cone-shaped PhoDAG-1 transformed into an almost cylindrical lipid, thereby decreasing the absolute value of J 0”

Add chemical structures of trans-PhoDAG-1 and cis-PhoDAG-1

Figures: 
What are the membrane composition? Only PhoDAG-1? Is this lipid inside domains or in the other phase? 
In general, the experiments are not clearly detailed. Add this information throughout the manuscript as well as the composition of each coexisting phases. Once defined, please explain why choising this and not other composition.

Table 1: indicate what is each parameter and where are the details of each calculation/determination. Regarding J calculation, if lipids mix, then J may differ markedly from the weighted sum. Therefore I think that this assumption is very naive.

Fig. 3:
“The same conclusion can be drawn from the slight decrease in the total domain number.”: please add quantification.

Line 195: what is the meaning of “inventory lipid”?

Line tension and lipid curvature has been the subject of the following papers, please take them into account:

Rufeil Fiori, E., Downing, R., Bossa, G. V., & May, S. (2020). Influence of spontaneous curvature on the line tension of phase-coexisting domains in a lipid monolayer: A Landau-Ginzburg model. The Journal of Chemical Physics, 152(5), 054707.

Bischof, A. A., & Wilke, N. (2012). Molecular determinants for the line tension of coexisting liquid phases in monolayers. Chemistry and physics of lipids, 165(7), 737-744.

Reviewer 2 Report

This manuscript is devoted to the physical chemistry of lipid domains and is an important step in understanding the fundamental laws of physicochemical biology. An international team of authors has done a great deal of theoretical and experimental work, the results of which are certainly of interest to a wide range of readers. However, before accepting the manuscript for publication, I would like to discuss a few points and comments.

1. It is extremely important to show the chemical formula of the lipids except typical like DPPC

2. I would like to know the opinion of the authors on the potential effect of cholesterol on the behavior of domains in the light of the hypotheses they consider

3. In Table 1 I would like to see the SD values. For many parameters, the difference for cis and trans columns is obvious. However, For example, how significant is the difference in the J0 parameter for LODs data?

The comments presented do not detract from the dignity of the work, which certainly represents a very interesting and topical study in the field of the physical chemistry of biomembranes.

Reviewer 3 Report

The manuscript entitled "Determinants of Lipid Domain Size" by Saitov et al. presents very interesting results, both experimental and computational, which enable verification of mechanisms regulating phase separation in lipid membranes. The authors verified two hypotheses, i.e. according to the first one, spontaneous monolayer curvature is the primary determinant of domain size, while the second one, prioritizes line tension. The presented research results are very solid and convincing, at the same time they indicate the correctness of the second hypothesis, where the factor determining the size of the domains is the line tension. In the opinion of the reviewer, the manuscript is suitable for publication, but some minor corrections are recommended.

  1. It is quite difficult for the reader to imagine the rate of growth of domains, the statements of „slow” or „fast” growth are quite vague. It would be worth quantifying it, e.g. by determining the increase in area per unit of time.
  2. In the text, the authors mention several types of energy under consideration. Therefore, it is a bit unclear what energy is exactly plotted in Figure 6. Is it total boundary energy?
  3. Due to the presence of a large azo group, it is somewhat questionable that the photolipid in the trans conformation has a cylindrical shape. What is the critical packing parameter value for this lipid? How does it change when switched to the cis conformation?
  4. The authors state that membrane quality was controlled by conductance and capacitance measurements. The values should be provided here.

Round 2

Reviewer 1 Report

My criticisms have been answered, the manuscript is now easier to follow. I have no further criticisms

Author Response

We thank the Reviewer for the positive evaluation of the manuscript. We did not introduce new changes since the Reviewer was satisfied with our reply to the criticism from the first round and did not raise new points in the second round.